# Remote physical function testing in older adults: a mixed methods study exploring test reliability, feasibility, and perceptions of participants and assessors

Jackson J. Fyfe[1*], Fernando Sousa[2], Kimberley Watson-Mackie[3], Paul Jansons[1,4], David Scott[1,4], Robin M. Daly[1]

1 Institute for Physical Activity and Nutrition (IPAN), School of Exercise and Nutrition Sciences, Deakin University, Geelong, Australia, 2 School of Primary and Allied Health Care, Monash University, Melbourne, Australia, 3 Faculty of Health, School of Health and Social Development, Deakin University, Burwood, Australia, 4 Department of Medicine, School of Clinical Sciences at Monash Health, Monash University, Clayton, Australia

* jackson.fyfe@deakin.edu.au

## Abstract

Impaired physical function is linked to poor health outcomes in older adults. Low-cost, remote strategies for monitoring physical function could enable timely interventions, improving health and reducing the need for support services. This study aimed to evaluate the feasibility, reliability, and perceptions of remotely assessing physical function in older adults at home. Nineteen community-dwelling older adults (68 ± 5 years; 68% female) participated in three videoconferencing sessions with trained assessors, one week apart. Participants completed nine physical function tests [standing balance test battery, single-leg balance, four-square step test (FSST), gait speed (2.44 m and 4 m, both usual and fastest pace), five-times sit-to-stand (5-STS), 30-second STS (30-STS)]. Semi-structured interviews with participants and assessors explored feasibility and acceptability. Test-retest reliability was assessed using intraclass correlation coefficients (ICCs) and Bland-Altman plots, and interview data were analysed thematically. All participants completed assessments without adverse events. Test-retest reliability ranged from poor (ICC < 0.5 for single-leg balance) to good (ICC 0.75–0.9 for 5-STS, 30-STS, gait, and FSST), improving with repeated testing. Participants identified challenges with device setup and physical demands of preparing their home. Assessors noted participant understanding as a barrier. Both groups recognised support from others during testing as a key enabler. Older adults can safely and feasibly perform physical function tests remotely. Familiarisation improves reliability and reducing technical and physical demands can enhance implementation. Support from others at home may be important for successful remote testing.

**Data availability statement:** The raw quantitative data and deidentified interview transcripts from this study are publicly available in the Open Science Framework (OSF) repository (https://doi.org/10.17605/OSF.IO/TH276).

**Funding:** The authors wish to thank the Institute for Physical Activity and Nutrition (IPAN), Deakin University for providing funding. The funders had no role in study design, data collection and analysis, decision to publish, or preparation of the manuscript.

**Competing interests:** The authors have declared that no competing interests exist.

## Introduction

Maintaining physical function with ageing is essential to perform activities of daily living [1]. Declines in physical function strongly predict health status and other important outcomes in older adults including disability, independence, falls risk, quality of life and even mortality [2, 3]. This signifies a need for preventive strategies that enable older people to maximise their physical function and maintain their independence [4].

The routine monitoring of physical function in older people could inform interventions to improve health and functional status and delay or reduce the need for residential care or community support services. Within research settings, assessment of physical function is often performed in older adults to monitor health status and the effectiveness of interventions. However, the monitoring of functional performance is often not part of routine clinical practice in older adults [5, 6], likely due in part to limited time and resources and a lack of awareness of poor physical function as a manageable health condition. The remote assessment of physical function in older adults via telehealth could overcome many of the barriers to in-person assessments and facilitate more routine monitoring to inform interventions to improve functional status and related outcomes.

Various common physical function tests including sit-to-stand, timed-up-and-go and gait speed assessments have established validity and reliability in controlled research settings in older adults [7, 8]. While several studies have examined the validity of these or similar tests performed remotely compared to in-person [9, 10, 11, 12, 13, 14], fewer have assessed the test-retest reliability of similar tests performed entirely remotely using digital technologies such as live videoconferencing [15,10,11,16,13]. A potential barrier to wider implementation of remote physical function tests in older adults is limited understanding of the feasibility and acceptability of performing such tests within the home environment. A recent systematic review [17] found measures of feasibility and acceptability were often not included in studies of remote physical function tests in older adults. Therefore, the aim of this mixed-methods study was to establish the test-retest reliability and feasibility of remote, home-based physical function testing in older adults, and the barriers and enablers of such testing from the perspectives of both participants and assessors.

## Methods

### Study design and setting

This was a convergent parallel mixed-methods study designed to assess the reliability and feasibility of validated physical function tests conducted remotely within the homes of community-dwelling older adults using videoconferencing technology. Participants attended a total of three videoconferencing sessions from within their homes approximately one week apart. Each session involved completion of the same battery of nine physical function tests. In addition, a semi-structured interview was conducted with both participants and assessors following the third testing session to further explore factors related to the feasibility and acceptability of performing the

tests remotely within their home environment. The study was conducted from November 2022 to January 2023. Ethics approval was obtained from the Deakin University Human Research Ethics Committee (HREC 2022-149) and all participants provided written informed consent.

## Participants and recruitment

Community-dwelling older adults aged 60−85 years were recruited Australia-wide via online (Facebook) advertising. Participants were deemed eligible to participate if they were: 1) English-speaking, 2) non-smoking, 3) able to walk unaided or with minimal assistance for ≥50 m, 4) cognitively intact as indicated by a score ≤2 on the Short Portable Mental Status Questionnaire (SPMSQ), and 5) had access to a computer, smartphone or tablet device with a stable network or internet/WiFi connection. Participants were excluded based on the following criteria: 1) acute or terminal illness likely to impact study involvement, 2) unstable or ongoing cardiovascular, metabolic, or respiratory disorders, 3) self-reported body mass index (BMI) ≥40 kg·m$^{-2}$, 4) musculoskeletal or neurological disorders impacting voluntary movement, 5) upper- or lower-extremity fracture in the past three months, or 6) inability to commit to the study and its requirements. The risk of participants experiencing an adverse event during exercise was determined using the Exercise and Sports Science Australia (ESSA) Adult Pre-exercise Screening System (APSS) [18]. Participants with signs or symptoms of unstable or unmanaged disease (i.e., if participants answered 'yes' to any of the Stage 1 questions of the ESSA APSS) were excluded. All participant screening procedures were completed during an initial videoconferencing session. All eligible participants provided verbal informed consent prior to participation in the study. To estimate anticipated intraclass correlation coefficient (ICC) values of 0.90 with a precision (95% confidence interval) of ± 0.20 (based on unpublished pilot data for the 5-STS), and a 80% probability of obtaining the desired precision, a total sample size of 19 was required [19].

## Test assessors

Two research assistants were recruited and upskilled by the research team to conduct the remote testing sessions with participants. One assessor was a qualified physiotherapist with previous experience in delivering exercise interventions remotely using digital technologies, and the other a trained exercise scientist with experience delivering exercise interventions in-person only. Both assessors were experienced with qualitative research methodologies. Each assessor completed all three assessments on the same participants assigned to them.

## Preparation and instructional videos

Following the screening session and prior to the first testing session, participants were sent (via email) a link to a short (~5 min) preparation video developed by the research team to help them prepare for their home-based assessment session. The preparation video contained information on how to prepare a location within their home to conduct the session, including ensuring the location had sufficient space for all tests (particularly the gait tests), setting up for each test [e.g., measuring distances for the gait speed tests, selecting an appropriate chair for the sit-to-stand (STS) tests], how to position their camera appropriately for each test, and suitable clothing and footwear for the session. Participants were asked to view the preparation video prior to the first testing session with the aim of minimising set-up time during the session. The preparation video deliberately contained no instructional information about any of the tests to be performed in the session to minimise the risk of participants practising the tests prior to the first session.

During the first testing session, participants were shown instructional videos developed by the research team for each physical function tests immediately prior to performing the tests. Thereafter, participants were given the option of being shown the instructional videos for each test during each subsequent testing session.

### Self-reported physical activity

Prior to undertaking the first testing session, participants were asked to complete (online via Qualtrics) the International Physical Activity Questionnaire -Elderly (IPAQ-E) to provide an estimate of physical activity and sitting time [20].

### Anthropometry and physical function assessments

Participants self-reported their height and weight to the nearest 1 cm and 1 kg, respectively. For the testing sessions, participants were required to have a > 4 m physical space available at home, a measuring tape (≥5 m), masking tape (or similar), and a suitable chair for sit-to-stand (STS) testing (approximately 43 cm in height, with a backrest, and without wheels or armrests). Participants were advised they could be provided with measuring and masking tapes if required. Participants undertook physical function tests during live videoconferencing sessions (via Zoom) on three occasions each separated by approximately one week. Sessions were conducted individually and delivered by one of the two research assistants. During each testing session, participants were required to have another person (e.g., family member or friend) present with them at their home to provide support if needed.

Physical function was assessed using the following battery of tests: 1) standing balance battery (side-to-side, semi-tandem and tandem); 2) single-leg balance test (with eyes open and closed); 3) four-square step test (FSST); 4) gait speed (2.44 m and 4 m, at both usual and fastest pace); 5) five-times STS (5-STS) test, and 5) 30-second STS (30-STS) test. For all tests, participants wore comfortable shoes or were barefoot, which was noted during the first testing session and repeated for subsequent testing sessions. The selected tests were designed to assess a broad range of physical function domains, including static balance (standing balance battery, single-leg balance test) dynamic balance and stepping speed (FSST), mobility (gait speed assessments), and lower-limb strength/strength-endurance (30-STS) and power (5-STS). Tests were completed in the same order for all participants and testing sessions. The time of day at which physical function testing was performed was not standardised between participants, but each testing session was conducted at a similar time of day for each participant where possible.

**Standing balance battery.** Participants' ability to maintain balance in three different stance positions (side-by-side, semi-tandem, and tandem) was assessed in accordance with the SPPB protocol [21]. The ability to maintain balance in the tandem stance position with the eyes closed was also assessed. Participants were asked to stand with their feet in full view of the camera (either on a smartphone, tablet, or webcam) and position themselves close to a wall or bench to provide support if needed. After assuming the correct stance position, participants were instructed they may use their arms, bend their knees, or move their body to maintain their balance, but to not move their feet. Balance time was recorded manually (in seconds) by the researcher using a stopwatch. Where balance could not be maintained for a given test, participants did not perform any further bilateral balance tests.

**Timed single-leg balance test.** Participants completed a timed single-leg balance test (on their preferred leg) of a maximum duration, both with the eyes open and separately with the eyes closed. A single practice trial (maximum of five seconds) was performed immediately before each test. Balance time was recorded manually (in seconds) by the researcher using a stopwatch. Participants were given a 3-second countdown, after which they assumed a single-leg stance and the timer was started. The timer was stopped when the participant's foot touched the ground or their any body part rested on another object.

**Sit-to-stand tests.** For the 5-STS and 30-STS tests, participants were asked to use a chair that: a) had a firm seat and backrest, b) had no armrests or wheels, and c) was at a height such that participants could place their feet flat on the floor while their upper body was in contact with the backrest. The same chair was used by each participant for both baseline and follow-up assessments. Before commencement of STS testing, participants were asked to position their chair side-on to the camera, and to adjust their camera so that their entire body was visible to the researcher when seated on the chair. If space limitations did not permit their entire body being visible on camera, the position of the chair and/or camera were adjusted so that at a minimum, the seat and backrest were within camera view.

For the 5-STS test, participants began from a seated position in the chair, with their arms folded across the chest, and were instructed to stand fully upright and then return to the seated position five times as quickly as possible. A single practice trial of the 5-STS was performed, with 1 min of passive recovery allowed before beginning the main test. The final score was recorded as the time taken (in seconds, to the nearest 0.01 seconds, recorded manually by the researcher using a stopwatch) to perform five STS repetitions from initially leaving the chair to being seated after the fifth repetition. The 30-STS test was then performed after a 5-minute recovery following completion of the 5-STS. For the 30-STS test, participants performed repeated chair stands in a manner identical to the 5-STS test; however, they were instructed to instead complete as many repetitions as possible in 30 seconds. The final score was recorded as the number of complete sit-to-stands (defined as standing in a fully upright position) achieved in 30 seconds. Participants were unable to view the 30-second timer during the test and were not provided with any feedback other than when to start and stop the test.

**Four-square step test (FSST).** The FSST was used to determine dynamic balance and stepping speed in four directions [22]. Participants were asked to step forwards, sideways and backwards over a cross formation marked on the floor using tape that was visible to the assessor. The test began with the participant moving first in a clockwise direction and returning in a counterclockwise position to the start square. Participants were asked to step with two feet into each square without touching or stepping on the tape as quickly as possible. A single practice trial was performed followed by one true attempt whereby the time of completing the sequence was recorded manually by the researcher in the nearest 0.01 of a second.

**Gait speed tests.** For the gait speed tests, participants were asked to walk both 2.44 and 4 m distances at both their usual and fastest pace. Participants were asked to measure each distance using a measuring tape and to place a household item at the start (i.e., at 0 m) and at both 2.44 and 4 m. Participants were asked to place their camera-enabled device approximately 2 m past the finish line in a diagonal (approximately 45°) direction. After confirming the participant was ready and in position at the start line, the researcher prompted the participant with "Ready, GO". The researcher began the stopwatch at "GO" and manually stopped the timer when the participant's centre of mass reached the finish line (either 2.44 m or 4 m). For the usual pace test, participants were asked to walk as they would normally as if they were not in a rush, and at their fastest possible pace for the other trial.

### Short physical performance battery (SPPB)

Results from the standing balance battery, gait speed (4 m usual pace) and 5-STS tests were used to calculate scores for the Short Physical Performance Battery (SPPB) [21]. For the standing balance battery, participants scored 1 point for each test in which balance was maintained for 10 seconds, and zero if balance was not maintained (for a maximum total score of 4 points). For the gait speed test (4 m usual pace), participants scored 1 point if their time was more than 8.70 seconds, 2 points if their time was between 6.21 and 8.70 seconds, 3 points if their time was between 4.82 and 6.20 seconds, and 4 points if their time was 4.81 seconds or less. For the 5-STS test, participants scored 0 points if they were unable to complete 5 chair stands or did so in more than 60 seconds, 1 point if their time was 16.70 seconds or more, 2 points if their time was 13.70 to 16.69 seconds, 3 points if their time was 11.20 to 13.69 seconds, or 4 points if their time was 11.19 seconds or less. The sum of the scores for each of tests was calculated as the total SPPB score.

### Feasibility of remote physical function assessments

The feasibility of undertaking home-based physical function testing remotely was assessed quantitatively by the proportion of participants who were able to successfully complete each test remotely during live videoconferencing sessions.

## Safety and adverse events

An adverse event was defined as any health-related unfavourable or unintended medical occurrence (sign, symptom, syndrome, illness) that developed or worsened during the period of observation in the trial. Any adverse events occurring during each testing session, or arising in the week following each testing session, were self-reported by participants in the next testing session or by contacting the research team via email or phone. Any reported adverse events were followed up by research staff to obtain further information and to advise whether participants should continue their participation in the study and/or seek medical advice.

## Interviews

Two members of the research team (FS and KW-M) conducted one-on-one semi-structured interviews (~15 min) with the participants via Zoom directly after the completion of the final testing session between October 2022 and January 2023. The interviewer of each participant was therefore the same person who conducted all physical function assessments with the interviewee. All participants in the quantitative feasibility study were invited to participate in interviews (convenience sampling) to ensure maximum variability sampling and richness of data. A separate member of the research team (JJF) conducted one-on-one semi-structured interviews with the two assessors who delivered the testing sessions with participants. The interview guide consisted of 10 open-ended questions (Supporting information file 1), followed up by additional probing questions from the interviewers as needed, to elicit thoughts and perceptions of both participants and assessors regarding the feasibility and acceptability of conducting physical tests in the homes of participants using videoconferencing. All interviews were audio-recorded and transcribed verbatim for subsequent analysis.

## Statistical analysis

**Quantitative data.** Participant characteristics are reported as means ± SDs for continuous variables and as number and percentages for categorical variables. Testing data are reported as either means ± SDs for continuous variables or median and interquartile range for categorical variables. Quantitative data relating to the feasibility of the home-based physical function tests was considered descriptive in nature. Test-retest reliability of physical function test scores was calculated between testing sessions 1 and 2, and sessions 2 and 3. Intraclass correlation coefficient (ICC) values (single-rating, absolute-agreement, 2-way random-effects model) were used to assess test-retest reliability. Interpretation of ICC values was based on classifications of poor (ICC < 0.5), moderate (0.5–0.75), good (0.75–0.9), or excellent (>0.9) [23]. Bland-Altman plots [24] of the difference between test-retest measurements (between trials 1 and 2, and trials 2 and 3) versus their mean values were generated for each test, including the mean difference and 95% CI, and 95% limits of agreement (estimated by mean difference ± 2 SD of the difference). Differences in physical function test scores across each testing session were also analysed using a one-way ANOVA. ICC data and one-way ANOVA were calculated using SPSS, and Bland-Altman plots were generated using GraphPad Prism (Version 9.5.1).

**Qualitative data.** Following each interview, the transcripts were sent to each participant for member checking [25]. Interview transcripts were then uploaded into NVivo software (Version 14, Lumivero, Denver, CO) for analysis. Analysis was conducted using qualitative description, a naturalist approach to data analysis that is considered an ideal method for mixed-methods research [26,25] as it focuses on identifying participants' perceptions of an intervention, and how an intervention might be improved and inform clinical practice [26,25]. Qualitative description is considered an ideal method for mixed-methods research investigating a health intervention as it seeks to understand the perceptions and experiences of the participants engaging in the intervention [26,25]. As qualitative descriptive methodology conducts data collection and analysis simultaneously, researchers began analyses of the interview transcripts while quantitative data was still being collected [26,25]. Using NVivo software, researchers coded the transcripts via line-by-line coding, initial coding, and focused coding using an inductive approach that kept the analysis in line with the participants experiences. The initial

coding of each transcript was performed by the same member of the research team who conducted the interview. Once focused coding was conducted, the research team discussed and condensed the codes to themes considered relevant to the research questions [26,25].

## Results

### Participant characteristics

A total of 26 older adults were screened for the study, of which 26 met the eligibility criteria. Seven participants withdrew from the study prior to the first testing session and after being sent the study information and instructional video, with the remaining participants (n = 19) completing all three remote functional assessments and the final interview.

Reasons for withdrawal included limited time availability (n = 3), misunderstanding that the study involved a training intervention (n = 2), a non-study-related injury (n = 1), and the perception that the testing would be too difficult (n = 1).

The characteristics of the 19 participants were (mean ± SD): age 68.5±4.9 (range 61–78) years; height 167.0±9.9 cm; weight 78.4±19.8 kg; body mass index (BMI) 27.9±4.9 kg·m$^{-2}$ [32% (n = 6) overweight, 37% (n = 7) obese]; weekly moderate-vigorous physical activity (MVPA) 584±606 min per week; and weekly sitting time 26.4±26.6 hours per week.

### Feasibility and safety of remote physical function assessments

All participants were able to successfully complete all physical function assessments on each of the three testing occasions. All participants had access to the required items for testing at home. No adverse events were self-reported by participants during the study.

### Test-retest reliability of physical function assessments

Mean ± SD (or median ± IQR) data for all physical function assessments undertaken for each of the three trials is shown in Table 1. The test-retest reliability of these assessments ranged from poor (ICC < 0.5; single-leg balance) to good (ICC 0.75–0.9; 5-STS, 30-STS, 2.44 m gait usual, 4 m gait fast, and FSST), and improved with repeated testing for most tests (i.e., reliability improved between testing session 2 and 3 compared to between testing session 1 and 2). Negative ICC values were observed for the single-leg balance test between test 1 and 2, suggesting the within-group variance

Table 1. Mean and SD (or median and interquartile range) for all physical function assessments performed for each of the three trials by the 19 participants. P-values reflect results from one-way repeated measures ANOVA assessing differences across trials.

| Test | Trial 1 | Trial 2 | Trial 3 | P-value |
|---|---|---|---|---|
| **5-STS** (sec)[1] | 15.10±2.80 | 14.19±2.76 | 14.20±2.16 | 0.030 |
| **30-STS** (number)[1] | 11.9±1.9 | 12.5±1.8 | 12.4±1.9 | 0.117 |
| **FSST** (sec)[1] | 15.10±2.80 | 14.19±2.76 | 14.20±2.16 | 0.016 |
| **Gait 2.44 m usual speed** (m/s)[1] | 0.57±0.14 | 0.66±0.20 | 0.64±0.18 | 0.079 |
| **Gait 2.44 m fastest speed** (m/s)[1] | 0.81±0.20 | 0.89±0.22 | 0.92±0.23 | 0.019 |
| **Gait 4 m usual speed** (m/s)[1] | 0.73±0.14 | 0.78±0.16 | 0.85±0.18 | 0.001 |
| **Gait 4 m fastest speed** (m/s)[1] | 1.01±0.20 | 1.10±0.26 | 1.14±0.27 | 0.065 |
| **Single-leg balance** (sec)[1] | 7.07±4.34 | 6.15±2.51 | 6.35±3.81 | 0.699 |
| **SPPB balance score** (number)[2] | 4±0 | 4±0 | 4±0 | 0.378 |
| **SPPB gait score** (number)[2] | 3±1 | 3±1 | 2±2 | 0.004 |
| **SPPB chair stand score** (number)[2] | 2±2 | 3±2 | 2±1 | 0.032 |
| **SPPB total score** (number)[2] | 9±2 | 10±2 | 10±2 | 0.002 |

[1]Values represent mean ± SD; [2]Values represent median ± IQR

exceeded between-group variance. As this violates a core assumption of ICC analysis, these results should be interpreted cautiously. A summary of test-retest reliability data for all physical function assessments is provided in Table 2. Bland-Altman plots for all tests are displayed in Fig 1 and Fig 2. Bias and limits of agreement values for all physical function assessments are shown in Supporting information file 2.

## Qualitative results

### Barriers

**Participants.**  Participants viewed the physical demands associated with preparing for the testing (including challenges associated with the need to move furniture, or place their camera on the floor), and technical issues as barriers to remote physical testing (Table 3). They mentioned space limitations for performing tests (Table 3, Q1), positioning the camera to capture the right angles (Table 3, Q2) and using materials to mark their floor (Table 3, Q3) as barriers to setting up. In addition, participants reported they had to move furniture around and perform some movements to set up that were perceived as physically demanding (Table 3 Q4-Q5). Picking their device up to watch the video demonstration and then putting it down to do each test was challenging for some participants (Table 3, Q7). Other technical issues were when participants couldn't join the videoconferencing (Zoom) meeting, their devices did not work appropriately, or they had trouble accessing links provided to them during the session (Table 3, Q6, Q8-Q9).

### Assessors

Assessors viewed participant understanding, personality, and technical or logistical issues as barriers to remote physical function testing in older adults (Table 4). The majority of participants seemed to not understand the purpose of completing

**Table 2.  Test-retest reliability and interpretation for all physical function assessments between trial 1 and trial 2, and between trial 2 and trial 3. Interpretation of intraclass correlation coefficient (ICC) values were based on classifications of poor (ICC < 0.5), moderate (0.5–0.75), good (0.75–0.9), or excellent (>0.9) [23].**

| Test | Trial comparison | Mean difference (95% CI) | ICC values | | | ICC interpretation | |
|---|---|---|---|---|---|---|---|
| | | | Point estimate | Lower 95% CI | Upper 95% CI | Point estimate | Range (95% CI) |
| 5-STS (s) | 1 vs. 2 | 0.91 (0.05, 1.77) | 0.76 | 0.45 | 0.90 | Good | Poor to Good |
| | 2 vs. 3 | 0.00 (−0.78, 0.77) | 0.80 | 0.55 | 0.92 | Good | Moderate to Excellent |
| 30-STS (number of stands) | 1 vs. 2 | −0.6 (−1.3, 0.1) | 0.67 | 0.33 | 0.86 | Moderate | Poor to Good |
| | 2 vs. 3 | 0.1 (−0.4, 0.6) | 0.83 | 0.61 | 0.93 | Good | Moderate to Excellent |
| FSST (s) | 1 vs. 2 | 1.21 (0.10, 2.32) | 0.66 | 0.29 | 0.86 | Moderate | Poor to Good |
| | 2 vs. 3 | 0.24 (−0.63, 1.11) | 0.80 | 0.54 | 0.92 | Good | Moderate to Excellent |
| Gait 2.44 m usual speed (m/s) | 1 vs. 2 | −0.91 (−0.18, −0.003) | 0.44 | 0.03 | 0.73 | Poor | Poor to Good |
| | 2 vs. 3 | 0.02 (−0.05, 0.08) | 0.78 | 0.52 | 0.91 | Good | Moderate to Excellent |
| Gait 2.44 m fastest speed (m/s) | 1 vs. 2 | −0.09 (−0.17, −0.003) | 0.63 | 0.26 | 0.84 | Moderate | Poor to Good |
| | 2 vs. 3 | −0.03 (−0.11, 0.05) | 0.73 | 0.43 | 0.89 | Moderate | Poor to Good |
| Gait 4 m usual speed (m/s) | 1 vs. 2 | −0.05 (−0.11, 0.005) | 0.65 | 0.29 | 0.85 | Moderate | Poor to Good |
| | 2 vs. 3 | −0.07 (−0.14, 0.003) | 0.56 | 0.17 | 0.80 | Moderate | Poor to Good |
| Gait 4 m fastest speed (m/s) | 1 vs. 2 | −0.09 (−0.15, −0.02) | 0.79 | 0.42 | 0.92 | Good | Poor to Excellent |
| | 2 vs. 3 | −0.04 (−0.16, 0.08) | 0.57 | 0.17 | 0.81 | Moderate | Poor to Good |
| Single-leg balance (s) | 1 vs. 2 | 0.92 (−1.54, 3.37) | −0.29 | −0.48 | 0.43 | Poor | Poor to Poor |
| | 2 vs. 3 | −0.20 (−1.72, 1.32) | 0.53 | 0.11 | 0.79 | Moderate | Poor to Good |
| SPPB total score (AU) | 1 vs. 2 | −0.8 (−1.3, −0.2) | 0.51 | 0.06 | 0.78 | Moderate | Poor to Good |
| | 2 vs. 3 | −0.1 (−0.5, 0.4) | 0.67 | 0.32 | 0.86 | Moderate | Poor to Good |

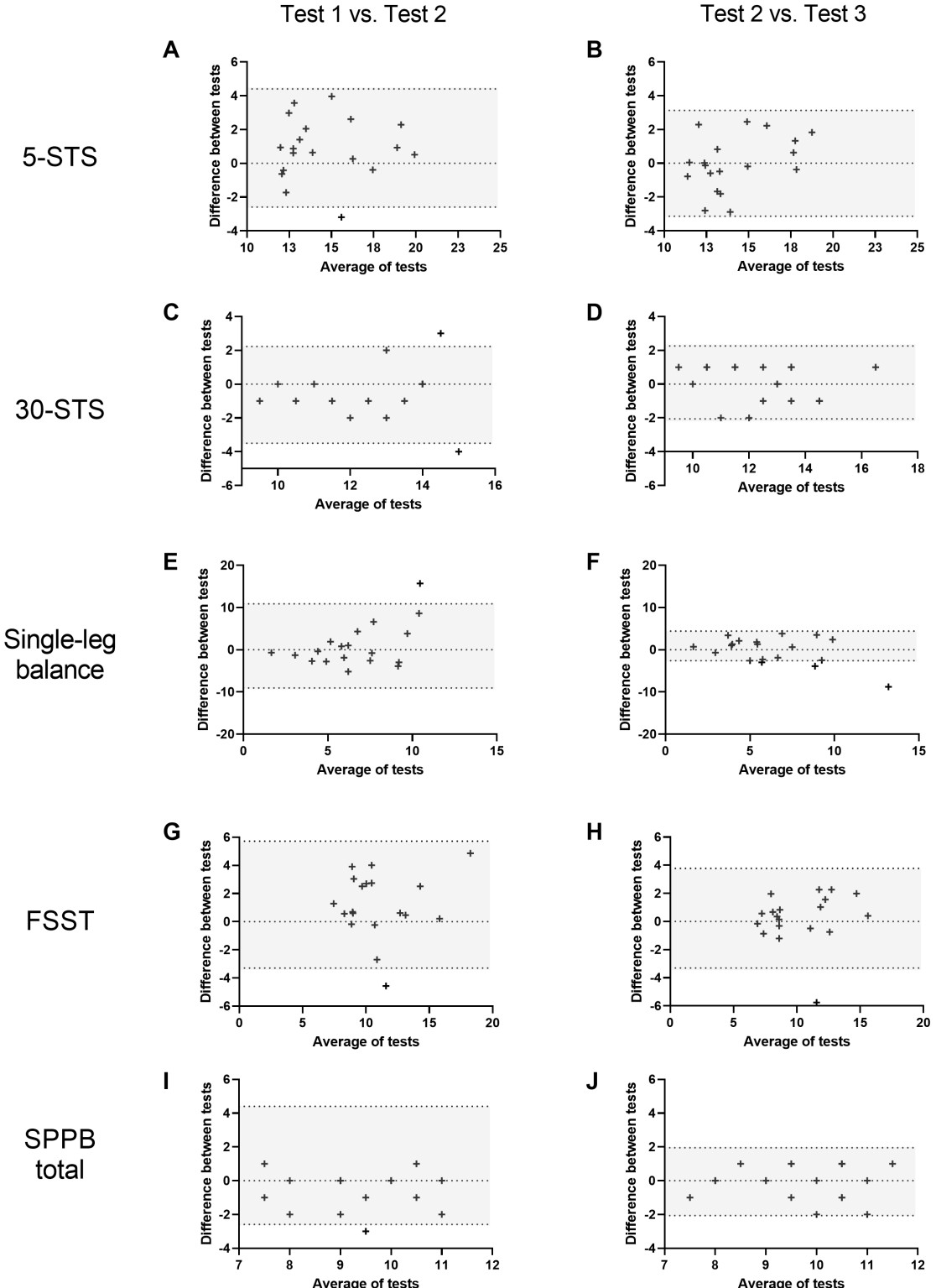

**Fig 1. Bland-Altman plots of differences in test scores between repeated testing occasions (session 1 minus session 2, and session 3 minus session 2) vs. averages of paired measurements for the five-times STS (5-STS) test (panels A and B), 30-second STS (30-STS) test (panels C and D), single-leg balance (panels E and F), four-square step test (FSST; panels G and H), and total score for the Short Physical Performance Battery (SPPB; panels I and J).** The dashed lines indicate the 95% confidence interval of the mean difference between trials. Shaded areas indicate 95% limits of agreement.

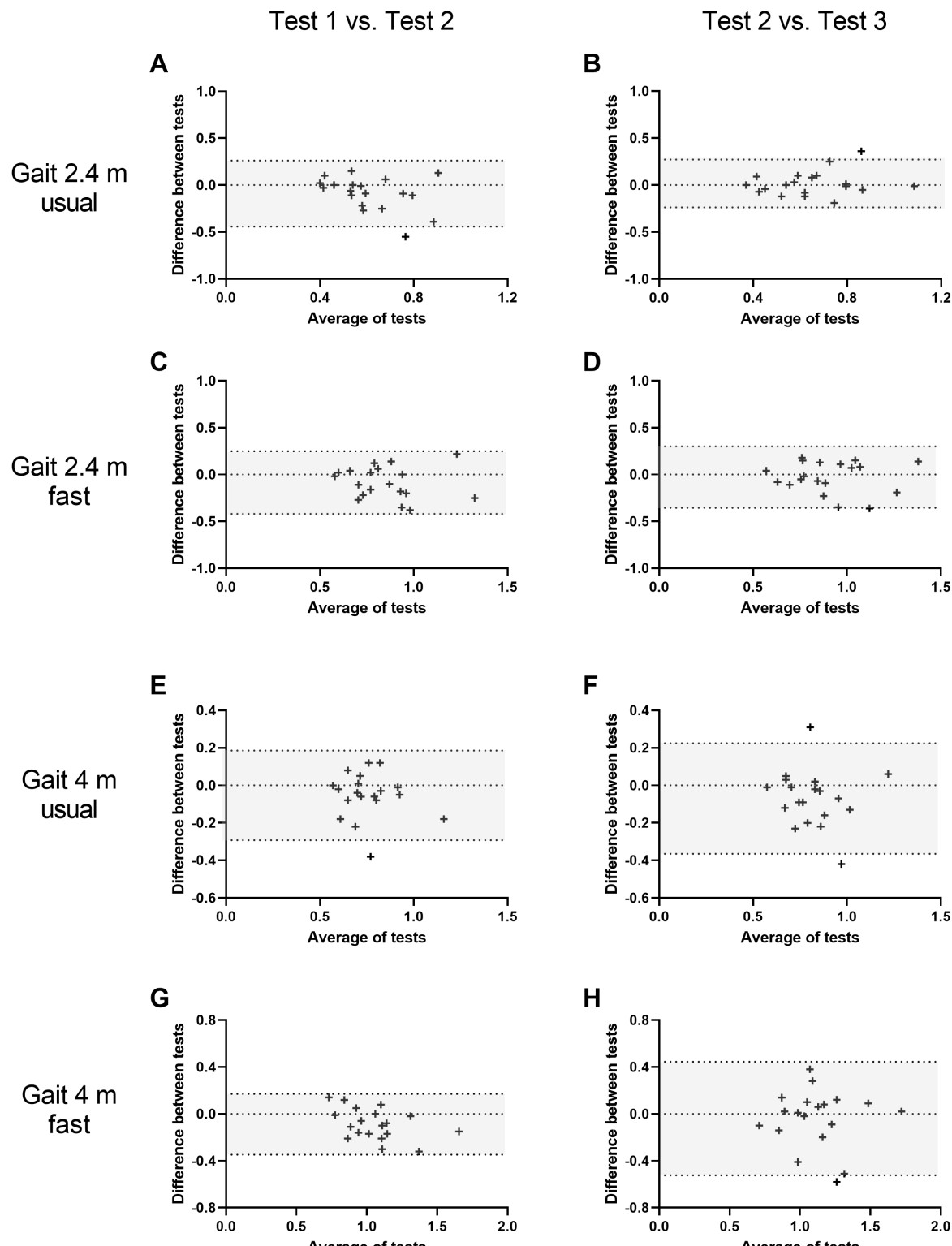

**Fig 2. Bland-Altman plots of differences in test scores between repeated testing occasions (session 1 minus session 2, and session 3 minus session 2) vs. averages of paired measurements for gait speed over 2.4 m at both usual (panels A and B) and fast speed (panels C and D), and gait speed over 4 m at both usual (panels E and F) and fast speed (panels G and H).** The dashed lines indicate the 95% confidence interval of the mean difference between trials. Shaded areas indicate 95% limits of agreement.

**Table 3. Barriers to remote physical function testing with older adults according to participants, including illustrative quotes.**

| Barriers | Quote | Illustrative Quote |
|---|---|---|
| **Setting up** | | |
| Physical space | Q1 | "I think because I live in a very small place it was challenging for me because I had a restricted area to do it. If I had a bigger—like on the video you showed the video, the instructor, he had a very large room. I think that would've been easier than my little room is very small and combined […] I was satisfied. The only thing that I wasn't happy about was my limited space. I felt that was very hard, restrictive for me." (Participant 1) |
| Camera positioning | Q2 | "The setting up the camera was the issue really. It wasn't the test itself, it was the camera. I used the computer, the laptop the first time and all the angles were too hard to manage, so I went to use the phone the second two times and that was easier, but trying to find a spot to put the camera." (Participant 10) |
| Marking the floor | Q3 | "As I flagged for you and I am never going to be putting masking tape onto a sealed marble floor. Having some options in mind for how to mark that grid. I've put a tissue and held on with the rubber band around the end of the tape measure because again, on a marble floor, I don't want the end of that metal piece on the end of the tape measure scratching the floor so just things I had to attend to." (Participant 3) |
| **Physical demand** | | |
| Moving furniture around | Q4 | "The first time I moved a lot of furniture around, it was like, "Oh no, I'll try that, and I'll try this." It took a long time. I said it was easy the second time and easy the third time, but trying to figure out the setup that was the hardest thing." (Participant 9) |
| Impairments of body function | Q5 | "An older person needing to set the camera up on the floor means they have to bend over and squat or sit on the floor and then get back up […] my knee's just a little bit unreliable and tender today. If it was playing up and swollen as it does get sometimes, then that would be an issue. That would probably be the more inconvenient part of it all." (Participant 10) |
| **Technical issues** | | |
| Connectivity problems | Q6 | "Like this morning when I couldn't connect, and that's what stressed me out more. The technology stressed me out more than the actual exercises didn't worry me at all." (Participant 15) |
| Picking the device up to watch the video for each test | Q7 | "The biggest problem is having to watch the videos and then respond and as I said to you with my eyesight and the size of a phone, there was no way that I was gonna manage to see a video if the phone was on the floor. If I have to keep picking it up, then that's really annoying actually 'cause there's whatever there is, there's 10 tests or something like that." (Participant 3) |
| Managing different links | Q8 | "I also would say sending me 10 different emails with links is okay if there's particular reason to do it that way. On the other hand, actually sending a link to a webpage, which has tasks one to 10 and would you please now click on task one and then we go from there […] I had to get out of YouTube, go back to the mail, open the next mail and so that's, again, it's another layer of technical, it's not complex, but it's areas where people could get lost." (Participant 3) |
| Device fault | Q9 | "Just my device, my computer was making it stop and start." (Participant 5) |

the physical function tests, particularly in a repeated manner (Table 4, Q1). Some participants also expressed some uncertainty in what they were expected to do during the testing sessions, which caused some degree of concern for the participants before their first session (Table 4, Q2). A common barrier was that some participants were unwilling to follow instructions or receive feedback where they felt confident about what they were required to do (Table 4, Q3). This was particularly true when instructions were repeated in the form of videos played before each test (Table 4, Q4). In some cases, participants showed a false sense of capability in performing the physical assessments, leading them to disregard safety-based instructions from assessors (Table 4, Q5).

## Enablers

### Participants

The instructional video sent to participants prior to the first testing session enabled the execution of the tests (Table 5, Q1), even though the physical space shown in the video seemed to be unrealistic to some participants (Table 5, Q1). The minimal and simple equipment required to set up was seen as an enabler for the tests (Table 5, Q2). Participants valued the visual instructions in the videos and the verbal instructions from assessors during the testing sessions (Table 5, Q3). Most participants mentioned the tests

**Table 4. Barriers to remote physical function testing with older adults according to assessors, including illustrative quotes.**

| Barriers | Quote | Illustrative Quote |
|---|---|---|
| **Participant understanding** | | |
| Poor knowledge of testing purpose | Q1 | "I think most participants didn't understand the purpose of the test. Sometimes, I had to explain at the end because they were not aware. As I said, most of them was thinking they were doing exercises. I felt like some of them were frustrated when we were repeating the tests, because as they were thinking it was exercises, they were expecting to do something different. Then they were like, 'Okay, why am I doing the same thing? Why am I not doing something different? How will I improve if I'm doing the same thing?' I think they didn't understand the purpose, and I don't know why." (Assessor 1) |
| Unclear requirements of testing | Q2 | They weren't sure what the tests were gonna involve, so going into the first session, they were like, 'I don't what I'm actually gonna be expected to do.' There was one participant who referred a few people that he knew, and he had said—he was interviewed by (other assessor), but in his transcripts, he said that he had been in contact with a lot—a lot of them had called him the night before the testing session, stressing about getting set up for the testing. Them saying like, 'What's the testing actually involve, like do I need weights?' 'Do I need this'? 'Do I need that'? (Assessor 2) |
| **Participant personality** | | |
| Variable responsiveness to instruction or feedback | Q3 | "I think what made it more challenging was the fact that some participants were so confident about what they need to do. For example, (I) had one participant that she didn't watch the video to the end, so in the middle of the video, she said, 'Okay, I already know what to do,' and she came behind, walking behind the—far from the camera and was placed in certain position, but it was not right. I was trying to correct her like, 'Okay, it is not this test. It's (the) other test,' but she was not listening like, 'Okay, I will do. I already know. I already know.' Sometimes it was challenging, like the participant who didn't listen (to) the instructions, like 'I know what to do, and I will do what I think I need to do.' (Assessor 1) |
| Dislike of repeated instruction | Q4 | "One participant, for example, didn't want to watch the videos in the second and in the third session because she said, 'I don't like seeing the videos because I have already watched it in the previous—in the first session, so to make it quicker, you just need to say what I need to do, and I will remember based on the video that I watched in the first time, and I will do because I already know what to do.'" (Assessor 1) |
| False sense of capability | Q5 | "Like, 'Okay, I know what I'm doing, so I know my—I have my sense of capability if I will fall or not. It's not you who will tell me if I will fall or not.' As I say, they are older adults. They are at home, so in certain situations, I felt like, okay I don't know how safe the participant is at this moment. You need to have that chair. I tried ask them multiple times, and they put the chair close, but I felt like a bit of hesitancy regarding this concept of false safety and the sense of capability." (Assessor 1) |
| **Technical or logistical issues** | | |
| Connectivity issues | Q6 | "When I was delivering some tests, I felt in some participants—for some participants, the internet connection was a bit slow, the video. It's like there was a delay in the video, so I don't know at which extent that delay in the video influenced my capacity of recording the exact time the participant have completed the test. I know, in some instances, it could impact the results of the test delivered online in comparison to the test delivered in person because in person you are seeing in real-time, so for assessing online, you need to make sure that the internet conditions are good and you can see appropriately in the right speed in the screen, you know?." (Assessor 1). |
| Camera positioning issues | Q7 | "A little bit, so there was a few where we were like—when we were getting set up, like figuring' where to put the camera, there was one woman who didn't have a—she had hers in a case, so she didn't realize that if she put it down, like leaning against something, the case would slide. Her first session went for almost an hour because every time she'd put the tablet down, it would stop." (Assessor 2) |
| Protocols overly complex | Q8 | "You have to access the email. You have to watch the video. You have to prepare the space. You have to have the measuring tape. Then you have to watch the video prior to the session. In multiple layers, maybe, of complexity for the participants. I felt like some participants were a bit hesitant about this. I had one participant had told to me like, 'Whoa, I didn't expect to do so many things.' (Assessor 1) |

were easy to do (Table 5, Q4) and encouragement from the assessor during the sessions was helpful (Table 5, Q5). External support from the family appeared to be a key enabler, particularly for resolving technical issues (Table 5, Q6).

## Assessors

Some participants reported positive attitudes towards home-based physical function testing, although others stated they would only do so regularly under medical instruction (Table 6, Q1). The opportunity to engage in conversation during the

**Table 5. Enablers of remote physical function testing with older adults according to participants, including illustrative quotes.**

| Enablers | Quote | Illustrative quote |
|---|---|---|
| Instructional video for setting up | Q1 | "I liked that there was the preparatory videos just showing particularly how to get the thing set up and actually having the video of the man demonstrating, so you knew exactly what was going on. I thought that needing information was good." (Participant 11) |
| Minimal equipment required | Q2 | "I required minimal things to set up; tape measure and tape on the floor, chair. It was very easy to do." (Participant 17) |
| Visual and verbal instructions | Q3 | "Having the researcher telling me what to do was fine, like having the videos for the first time, and then he just reminded me what each one was as we went through that made it easier, because we got through it quicker and that was fine by me." (Participant 9) |
| Tests were easy | Q4 | "I found it pretty easy to do the tests and all that sort of thing." (Participant 16) |
| Practitioner features | Q5 | "The practitioner has been very encouraging and, yeah, clear in your instructions. It's good." (Participant 7) |
| External support | Q6 | "I'm lucky I've got backup to set up the Zoom for me and that, so at the beginning, I needed some help from my son to set all that up." (Participant 15) |

testing sessions helped build rapport between assessors and participants (Table 6, Q2). The assessors reported that being allowed some flexibility in how they delivered the testing protocols to participants (for example, only showing the instructional videos on repeated occasions when deemed necessary) enabled the testing sessions flow more smoothly (Table 6, Q3). Although not a study requirement per se, participants having access to a second camera-enabled device allowed testing sessions to be implemented more easily. For example, having two camera-enabled devices available meant participants could leave one device in place during the entire session for assessors to view them while completing each test, and use the second device to watch instructional videos as required (Table 6, Q4). Lastly, the assessors mentioned the availability of external support (e.g., family members) during the testing sessions alleviated difficulties some participants experienced with regards to technology (e.g., accessing Zoom link) and other logistical factors (e.g., setting up equipment and positioning camera appropriately) (Table 6, Q5).

**Table 6. Enablers of remote physical function testing with older adults according to assessors, including illustrative quotes.**

| Enablers | Quote | Illustrative quote |
|---|---|---|
| Variable participant attitudes towards testing | Q1 | "I think they would, but I think a few of them thought of it as like exercises that you would do every day, and then other ones were like they'll use it to self-assess, and then other ones were like, 'I'd if a doctor told me to.' Beyond that, they were just like, 'I don't really see the point. Unless a doctor's doing it for me, tellin' me that I need to do it, then I don't really see the point.' It really varied, so some people were like, 'I'm going to do these exercises every day,' and then other ones were like, 'Yeah, maybe,' and then others were just like, 'No.' It really did vary a lot." |
| Rapport between assessors and participants | Q2 | "The others tried to engage with me like trying to talking about—random talks during the session, so they were very talkative, and they tried to engage with me, asking questions, personal questions about me, for example. I felt like we had some bond being built during that short sessions, and it was good." (Assessor 1) |
| Partial flexibility in protocols | Q3 | "Having that flexibility built in, like it became a lot easier once I knew that it was like I could read the person and what the person wanted and act accordingly with it. That was very helpful." (Assessor 2) |
| Access to multiple devices with camera | Q4 | If it could be sort of in place where they were able to have a second device all ready to go,'cause everyone pretty much agreed the second device is easier. There was one person who didn't have a second device, so that's probably something else that has to be considered as well. (Assessor 2) |
| External support during testing | Q5 | "I think it's important to mention that some participants had extra support from the family. For example, some participants had difficulty on join the Zoom link. One participant mentioned to me that she asked to his son to set up everything and instruct her how to access the Zoom, open the calendar with the link, so yeah, I think. The other participant had the husband at home on her side during the test as a backup solution, like "If I have problems, you are here, and you will help me," so I think it was important to them." (Assessor 1) |

## Discussion

The main findings of this study were i) remote assessment of common physical function assessments were feasible and could be performed safely within the homes of older adults living independently in the community; ii) the test-retest reliability of these assessments ranged from poor (ICC < 0.5; single-leg balance) to good (ICC 0.75–0.9; 5-STS, 30-STS, 2.44 m gait usual, 4 m gait fast, and FSST) and improved with repeated testing for most tests; iii) common barriers to remote physical function testing from the perspectives of participants included the physical and technical demands associated with preparing to complete the tests remotely, while assessors saw participant understanding and receptiveness as barriers, and iv) both participants and assessors identified the availability of support people during testing as enablers.

### Test-retest reliability and comparison to other studies with remote-only tests

The test-retest reliability of the remote physical function tests assessed in this study varied considerably, with ICC scores ranging from poor (i.e., single-leg balance in test 1 versus test 2) to moderate (i.e., 2.44 m gait fast, 4 m gait usual, and SPPB total score) or good (i.e., 5-STS, 30-STS, 2.44 m gait usual, 4 m gait fast, and FSST). Of note were the negative ICC values observed for the single-leg balance test between test 1 and 2 (but not between test 2 and 3), suggesting these data should be interpreted with caution. In general, we found better test-retest reliability with sit-to-stand (5-STS and 30-STS), dynamic mobility (FSST) and short distance (2.44 m) gait tests than with longer-distance (4 m) gait speed or balance tests. While several studies have examined the test-retest reliability of in-person physical function assessments [7, 8], or the validity of tests performed remotely compared to in-person [9, 10, 11, 12, 13, 14], fewer have assessed the test-retest reliability of similar tests performed entirely remotely using digital technologies such as live videoconferencing [15, 11]. One study examined the test-retest reliability of physical function assessments delivered via telehealth and agreement with in-person assessments in people (aged ≥45 years) with chronic lower-limb musculoskeletal disorders [11]. Several tests (stair climb test, timed up-and-go, right leg timed single-leg stance, and calf raises) were identified as having good-to-excellent test-retest reliability (ICC = 0.84–0.91) with an acceptable lower 95% CI defined as above 0.7 [11]. Other tests including the 30-STS, 5-m fast walk, step test, and left leg timed single-leg stance showing moderate-to-good reliability (ICC = 0.69–0.81), but without acceptable lower 95% CI agreement [11]. Of the physical function tests used in this previous study [11], we found poorer reliability for the single-leg balance test (conducted on the preferred leg only), particularly between the first two testing sessions (ICC = −0.29), although this improved but was classified as moderate between sessions two and three (ICC = 0.53 in the present study versus 0.69–0.84 for left and right legs [11]). For the 30-STS test, we found comparable moderate-to-good reliability (ICC 0.67 to 0.83) to that reported by Lawford et al. (ICC = 0.77) [11]. These ICC values were however lower than reported in other studies assessing the reliability of the 30-STS test (ICC = 0.95) conducted via telehealth in younger healthy individuals [15]. Although we assessed gait speed over shorter distances than previous studies (2.4 m or 4 m versus 5 m in other studies [11]), we found similar reliability for the 4-m fast walk (ICC = 0.75) than seen over longer distances (ICC = 0.71) [11].

Overall, we found generally lower ICC scores for most tests than previous studies that have examined the reliability of remote-only assessments [15, 11]. Like previous studies [15, 11], we used a pragmatic approach whereby participants conducted testing within their own home environment, set-up the tests themselves using readily-available equipment, and the tests were assessed using a freely-available videoconferencing platform accessed on any suitable device. While these factors improve the utility of this pragmatic approach in clinical and research settings [11], this may have increased the variability between test scores on repeated occasions due to errors in measuring distances, possible lags in live footage due to connection issues, and differences in camera positioning between tests.

### Effect of repeated testing and familiarisation on test-retest reliability

Another key finding was that the test-retest reliability of most physical function tests improved with repeated testing (i.e., ICC scores were typically higher between T2-T3 than for T1-T2). This is in line with observations from laboratory-based testing [27] and underscores the importance of familiarising older adults to remote physical function tests prior to

completing baseline or follow-up assessments in clinical or research settings. Previous studies [15, 11] assessing the reliability of remote physical function testing have generally not included a familiarisation testing session prior to the main test-retest reliability trials. While these studies [15, 11] may have underestimated the true reliability of physical function assessments as a result, they provide an indication of the reliability of these assessments without prior familiarisation, which may not always be feasible in practice. Nevertheless, our findings highlight the importance of test familiarisation in improving the test-retest reliability of common physical function assessments when conducted remotely in older adults, and in turn, their sensitivity for detecting changes in physical performance over time.

### Test feasibility and safety

Another important finding from our study was that community-dwelling older adults can feasibly and safely perform physical function tests within their home environment during live videoconferencing sessions with an assessor. This was supported by all 152 tests being successfully completed by participants across the three testing sessions, with no adverse events reported. A 2023 systematic review [17] found only 3 of 17 available studies involving the remote collection of physical performance measures undertook quantitative assessment of the feasibility or acceptability of such testing. While the findings of these studies were generally positive regarding feasibility of remote physical function testing, the review concluded there was insufficient evidence to draw conclusions regarding the feasibility or acceptability of these measures when collected remotely [17]. The present study therefore adds to our current understanding of the feasibility and safety of remote home-based physical function testing and suggests older adults with comparable physical function levels to the participants in this study can feasibly complete these assessments remotely within their home environment when monitored by a trained assessor. Nevertheless, the fact that four of the seven participants who withdrew prior to the first testing session cited limited time availability ($n = 3$) and perceived testing difficulty ($n = 1$) suggests that shortening or simplifying the testing battery could further enhance feasibility. One strategy may be to omit tests with lower reliability or those assessing overlapping domains of physical function. For example, the single-leg balance test demonstrated poor test-retest reliability between the first two sessions (ICC = −0.29), with only moderate reliability between sessions two and three (ICC = 0.53). This is notably lower than the reliability reported in similar telehealth studies using single-leg balance tests (ICC = 0.69–0.84) [11], and indicates this test may contribute unnecessary burden without providing robust or consistent data. Similarly, tests that capture similar constructs - such as the 4-m and 2.44-m gait speed assessments - could be streamlined, especially given the similar reliability observed for the 2.44-m fast walk (ICC = 0.73) and 4-m fast walk (ICC = 0.79). Retaining higher-performing and less redundant tests, such as the 5-STS, 30-STS, and FSST, which consistently showed moderate-to-good reliability, may streamline the testing protocol without compromising measurement quality.

### Barriers and enablers to remote physical function testing

A novel aspect of this study was the identification of barriers and enablers to home-based physical function testing from the perspectives of both participants and assessors. Some of the major barriers faced by participants related to logistical factors such as setting up and operating technology during each session. Although the testing required minimal equipment, some participants had to move furniture to prepare a space to perform the tests within their home, which they perceived as physically demanding. They also found it challenging to watch instructional videos on the same device they were using for videoconferencing, as this required them to repeatedly pick up and reposition the device during the session. The assessors found the unwillingness of some participants to follow test instructions to be a major barrier to testing. This was particularly evident when participants believed they were capable and knew how to perform the tests. Some participants became less receptive to instruction over successive testing sessions. It also became evident to assessors that some participants had limited understanding of the purpose of the tests (for example, as assessments of their physical function versus an intervention to improve their physical function), despite being provided with prior information regarding the purpose of the tests. Another key barrier highlighted by both participants and assessors were difficulties with

using technology during the testing sessions, which in some cases required the assistance of a support person who was present with the participant. These observations together suggest minimising the set-up and technological requirements of testing, and avoiding repeated instruction where appropriate, may improve the feasibility of remote home-based physical function testing in older adults.

Many participants found the pragmatic nature of the testing protocols, specifically that the tests were easy to perform and required minimal equipment, to be key enablers. They also found the instructional video provided to them prior to the first testing session helpful for them to prepare. The assessors reported most participants had positive attitudes towards home-based physical function testing. Despite this, many reported they would only undertake regular physical function assessments under medical instruction, which highlights the critical role of healthcare professionals in promoting interventions to monitor and maintain physical function across the life span. Factors related to technology also enabled testing. Given the requirement for participants to watch standardised videos of the testing protocols, having two camera-enabled devices allowed participants to leave one device in place during the entire session for assessors to view them while completing each test, and use the second device to watch instructional videos as required. External support from the family was a key enabler from the perspectives of both participants and assessors, particularly for resolving technical issues. These factors could be leveraged in future research or community interventions to monitor physical function remotely in older adults.

## Strengths

This is one of only a few investigations to explore, in addition to measures such as test-retest reliability, factors related to the feasibility and/or acceptability of remote physical function testing in community-dwelling older adults. The inclusion of qualitative data from both participants and assessors provides new insights that may inform future implementation strategies for the remote monitoring of physical function in older adults. We also assessed the test-retest reliability of remote physical function tests over multiple (i.e., three) sessions, which allowed potential learning effects associated with repeated testing to be quantified, highlighting the importance of test familiarisation to improve measurement reliability.

## Limitations

The physical function levels of the older adults in this study may have influenced the feasibility and/or reliability of the physical function measures. Individuals with SPPB scores of 10 or less have a significantly higher risk of mobility disability [28] and gait speeds of <1.0 m/s [29] or 0.8 m/s [30] are recommended for identifying those at risk of disability. Participants in this study had a median SPPB score of 9 (17/19 participants scored 10 or less; Supporting information file 3) and usual gait speeds of 0.73 m/s (in trial 1) for the 4 m test. While this suggests representation from older adults with reasonably poor physical function, despite the relatively high (albeit variable) weekly MVPA levels across the cohort (584 ± 606 min per week), which is often overestimated in self-reported MVPA questionnaires [31], those with lower physical function may experience greater challenges completing physical function tests remotely. We recruited the participants in this study using online advertising, which may have biased our sampling towards older adults with higher digital literacy and in turn positively influenced the feasibility of testing. Although this study focused on the reliability of remote physical function assessments, we did not evaluate their agreement with in-person measures. A 2023 systematic review [17] found six of nine studies reported good agreement between remote and face-to-face assessments (<5% differences between measures), although none showed consistently good agreement across studies. It is therefore important to confirm whether the specific physical function tests in this study show acceptable agreement with the equivalent face-to-face measures. We also did not evaluate inter-rater reliability, although this would have required both a larger sample and different ICC methodology. Conducting interviews only after the third testing session may have introduced recency bias, where participants' responses were disproportionately influenced by their most recent experience, such as any technical difficulties encountered, which, by that stage, had been largely resolved for all participants. Finally, the use of the same assessor

to conduct the physical function assessment and interviews with each participant, which was a pragmatic decision to limit participant burden (given interviews were completed immediately after the final testing session) may have biased their responses.

## Conclusions

Community-dwelling older adults can feasibly and safely perform a range of physical function tests within their home environment. Although test reliability ranged from poor (ICC < 0.5; single-leg balance) to good (ICC 0.75–0.9; 5-STS, 30-STS, 2.44 m gait usual, 2.44 m gait fast, and FSST), this improved with prior familiarisation. Limiting the physical and technical demands associated with testing, and leveraging the availability of support people at home, may facilitate the wider implementation of such testing in older people for research or clinical purposes.

## Supporting information

**S1 File.** Interview schedule.
(DOCX)

**S2 File.** Bias and limits of agreement values for all physical function assessments between tests 1 versus 2 and between tests 2 versus 3.
(DOCX)

**S File.** Frequencies of total SPPB score across the participant cohort.
(DOCX)

## Author contributions

**Conceptualization:** Jackson J. Fyfe, Paul Jansons, David Scott, Robin M. Daly.

**Formal analysis:** Jackson J. Fyfe, Fernando Sousa, Kimberley Watson-Mackie, Robin M. Daly.

**Funding acquisition:** Jackson J. Fyfe, Paul Jansons, David Scott, Robin M. Daly.

**Investigation:** Jackson J. Fyfe, Fernando Sousa, Kimberley Watson-Mackie.

**Methodology:** Jackson J. Fyfe, Fernando Sousa, Kimberley Watson-Mackie, Paul Jansons, David Scott, Robin M. Daly.

**Project administration:** Jackson J. Fyfe.

**Supervision:** Jackson J. Fyfe.

**Visualization:** Jackson J. Fyfe.

**Writing – original draft:** Jackson J. Fyfe.

**Writing – review & editing:** Jackson J. Fyfe, Fernando Sousa, Kimberley Watson-Mackie, Paul Jansons, David Scott, Robin M. Daly.

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
