## [Decision Letter · Decision Letter 0]

19 Jun 2025

PONE-D-25-19949Remote physical function testing in older adults: a mixed methods study exploring test reliability, feasibility, and perceptions of participants and assessors.PLOS ONE

Dear Dr. Fyfe,

Thank you for submitting your manuscript to PLOS ONE. After careful consideration, we feel that it has merit but does not fully meet PLOS ONE’s publication criteria as it currently stands. Therefore, we invite you to submit a revised version of the manuscript that addresses the points raised during the review process.

**Please address the reviewers' comments thoroughly. For any concerns that can be reasonably resolved, please revise the manuscript accordingly. If certain issues cannot be addressed, kindly provide a clear and well-justified explanation in your response.**

We look forward to receiving your revised manuscript.

Kind regards,

Jindong Chang, Ph.D.

Academic Editor

PLOS ONE

**Journal Requirements:**

1. When submitting your revision, we need you to address these additional requirements. Please ensure that your manuscript meets PLOS ONE's style requirements, including those for file naming. The PLOS ONE style templates can be found at https://journals.plos.org/plosone/s/file?id=wjVg/PLOSOne_formatting_sample_main_body.pdf and https://journals.plos.org/plosone/s/file?id=ba62/PLOSOne_formatting_sample_title_authors_affiliations.pdf 2. Thank you for stating the following financial disclosure: The authors wish to thank the Institute for Physical Activity and Nutrition (IPAN), Deakin University for providing funding.   Please state what role the funders took in the study.  If the funders had no role, please state: "The funders had no role in study design, data collection and analysis, decision to publish, or preparation of the manuscript." If this statement is not correct you must amend it as needed. Please include this amended Role of Funder statement in your cover letter; we will change the online submission form on your behalf. 3. Thank you for stating the following in the Acknowledgments Section of your manuscript: The authors wish to thank the participants for their efforts during the study and the Institute for Physical Activity and Nutrition (IPAN), Deakin University for providing funding. We note that you have provided funding information that is not currently declared in your Funding Statement. However, funding information should not appear in the Acknowledgments section or other areas of your manuscript. We will only publish funding information present in the Funding Statement section of the online submission form. Please remove any funding-related text from the manuscript and let us know how you would like to update your Funding Statement. Currently, your Funding Statement reads as follows: The authors wish to thank the Institute for Physical Activity and Nutrition (IPAN), Deakin University for providing funding.  Please include your amended statements within your cover letter; we will change the online submission form on your behalf. 4. In this instance it seems there may be acceptable restrictions in place that prevent the public sharing of your minimal data. However, in line with our goal of ensuring long-term data availability to all interested researchers, PLOS’ Data Policy states that authors cannot be the sole named individuals responsible for ensuring data access (http://journals.plos.org/plosone/s/data-availability#loc-acceptable-data-sharing-methods). Data requests to a non-author institutional point of contact, such as a data access or ethics committee, helps guarantee long term stability and availability of data. Providing interested researchers with a durable point of contact ensures data will be accessible even if an author changes email addresses, institutions, or becomes unavailable to answer requests. Before we proceed with your manuscript, please also provide non-author contact information (phone/email/hyperlink) for a data access committee, ethics committee, or other institutional body to which data requests may be sent. If no institutional body is available to respond to requests for your minimal data, please consider if there any institutional representatives who did not collaborate in the study, and are not listed as authors on the manuscript, who would be able to hold the data and respond to external requests for data access? If so, please provide their contact information (i.e., email address). Please also provide details on how you will ensure persistent or long-term data storage and availability. 5. We notice that your supplementary figure 1, table 1 and file 1 are included in the manuscript file. Please remove them and upload them with the file type 'Supporting Information'. Please ensure that each Supporting Information file has a legend listed in the manuscript after the references list. 6. We note that this data set consists of interview transcripts. Can you please confirm that all participants gave consent for interview transcript to be published? If they DID provide consent for these transcripts to be published, please also confirm that the transcripts do not contain any potentially identifying information (or let us know if the participants consented to having their personal details published and made publicly available). We consider the following details to be identifying information:- Names, nicknames, and initials- Age more specific than round numbers- GPS coordinates, physical addresses, IP addresses, email addresses- Information in small sample sizes (e.g. 40 students from X class in X year at X university)- Specific dates (e.g. visit dates, interview dates)- ID numbers Or, if the participants DID NOT provide consent for these transcripts to be published:- Provide a de-identified version of the data or excerpts of interview responses- Provide information regarding how these transcripts can be accessed by researchers who meet the criteria for access to confidential data, including:a) the grounds for restrictionb) the name of the ethics committee, Institutional Review Board, or third-party organization that is imposing sharing restrictions on the datac) a non-author, institutional point of contact that is able to field data access queries, in the interest of maintaining long-term data accessibility.d) Any relevant data set names, URLs, DOIs, etc. that an independent researcher would need in order to request your minimal data set. For further information on sharing data that contains sensitive participant information, please see: https://journals.plos.org/plosone/s/data-availability#loc-human-research-participant-data-and-other-sensitive-data If there are ethical, legal, or third-party restrictions upon your dataset, you must provide all of the following details (https://journals.plos.org/plosone/s/data-availability#loc-acceptable-data-access-restrictions):a) A complete description of the datasetb) The nature of the restrictions upon the data (ethical, legal, or owned by a third party) and the reasoning behind themc) The full name of the body imposing the restrictions upon your dataset (ethics committee, institution, data access committee, etc)d) If the data are owned by a third party, confirmation of whether the authors received any special privileges in accessing the data that other researchers would not havee) Direct, non-author contact information (preferably email) for the body imposing the restrictions upon the data, to which data access requests can be sent

Reviewers' comments:

Reviewer's Responses to Questions

**Comments to the Author**

1. Is the manuscript technically sound, and do the data support the conclusions?

Reviewer #1: Yes

2. Has the statistical analysis been performed appropriately and rigorously? 

Reviewer #1: I Don't Know

3. Have the authors made all data underlying the findings in their manuscript fully available?

Reviewer #1: Yes

4. Is the manuscript presented in an intelligible fashion and written in standard English?

Reviewer #1: Yes

5. Review Comments to the Author

**Reviewer #1: ** The present manuscript describes an investigation into the reliability of remote physical function tests, repeated three times, in older adults. The data provide quantitative scores of reliability along with a qualitative description of key enablers and barriers for remote testing from the participant and assessor view point.

Overall, this is a well written manuscript with a clear focus that will add to the evidence base around remote assessment of physical function of older adults, however there are some key issues that need to be addressed before publication can be recommended.

Please note, I returned the answer ‘I don’t know’ to the Review Question ‘Has the statistical analysis been performed appropriately and rigorously?’ specifically because more information is required regarding the ICC model used. The overall approach of using ICC to assess reliability in this context appears sound to me.

Reviewer’s comments:

Introduction:

Lines 64-66: A slightly pernickety point but given the literature available and described in the rest of the paragraph, it is probably not appropriate to say that understanding of the validity of remote vs in-person testing is particularly limited anymore- validity in this context is becoming reasonably well established now.

More rationale could be provided on the choice of functional tests, and this might inform a discussion point on which tests could be dropped from such a remote functional testing battery in future iterations, particularly in the context of the barriers related to set up etc.

Methods:

Line 158: ‘all participants had access to these items at home’ is really a result that should be reported in the feasibility section not methods.

Lines 188-195: were the single leg balance tests attempted for a maximum duration with no time cap?

Lines 297-298: please specify which ICC model was used, see 10.1016/j.jcm.2016.02.012 for further details and rationale for the importance of this.

Statistical analysis: to give a complete picture of the data, the data presented in Table 1 could be compared by repeated measures ANOVA / GLM to see if there were differences in the mean scores of each test across time.

Results:

Line 333: were reasons for drop-out provided, and could these be considered in the context of feasibility? On reading the qualitative results, were these drop-outs after receiving instructions for room set up? If so, this is an important consideration for feasibility?

Line 339: Weekly MVPA seems high for this age group- is that worth mentioning in the discussion?

Lines 348-352: this feels like it is missing written results giving an objective summary of the key data in the tables and figures. Given the data is quite dry and there is a lot of it, highlighting the headlines that are later discussed is all that is needed. For readability, these written results might be best broken up to proceed the respective table or figure.

Table 2: the negative ICC value might be indicating that there is a problem with the data, or the ICC model used is not appropriate. If these have been checked, then generally a negative ICC should not be meaningfully interpreted as the within-group variance is greater than the between-group variance, which contradicts one of the assumptions needed for ICC calculation.

Figure 1: should each panel be individually labelled and referred to in the figure legend? Also, for ease of interpretation, the axis formatting for test 1 vs test 2, and test 2 vs test 3, should be the same- this is different for 30-STS, FSST, and SPPB scores.

Figure 2: axis alignment is a bit off for some of the panels

Layout of Qualitative Results section: it might be a quirk of the submission process, but readability would be improved if the written results for given assessment of feasibility preceded the corresponding table of quotes, i.e., summary of participant’s enablers followed by table of participant’s enabler quotes, then summary of assessor’s enablers followed by table of assessor’s enabler quotes.

Discussion:

- Given the interview was after third assessment only, is there risk of recency bias, for example when considering technical difficulty? After three sessions, technical issues of the first session may be downplayed.

- Possibly linking back to a comment on the introduction regarding rationale for inclusion of each test, providing pragmatic suggestions for how these data might inform future implementation (alluded to in lines 585-587) would be an interesting and useful addition to the manuscript.

- A limitation that ought to be mentioned was that no comparison of reliability between assessors was made, though that would have required a larger sample size and different ICC model.

6. PLOS authors have the option to publish the peer review history of their article (what does this mean? ). If published, this will include your full peer review and any attached files.

**Do you want your identity to be public for this peer review?** For information about this choice, including consent withdrawal, please see our Privacy Policy .

Reviewer #1: No

---

## [Author Response · Author response to Decision Letter 1]

24 Aug 2025

Response to reviewers

Reviewer #1: The present manuscript describes an investigation into the reliability of remote physical function tests, repeated three times, in older adults. The data provide quantitative scores of reliability along with a qualitative description of key enablers and barriers for remote testing from the participant and assessor view point.

Overall, this is a well written manuscript with a clear focus that will add to the evidence base around remote assessment of physical function of older adults, however there are some key issues that need to be addressed before publication can be recommended.

Please note, I returned the answer ‘I don’t know’ to the Review Question ‘Has the statistical analysis been performed appropriately and rigorously?’ specifically because more information is required regarding the ICC model used. The overall approach of using ICC to assess reliability in this context appears sound to me.

Response: We would like to thank the reviewer for their time and expertise and for providing a number of useful suggestions on our manuscript. We have now made numerous changes based on the comments made and believe the manuscript has been improved significantly as a result. We have detailed our responses to each of the queries raised and highlighted where specific changes have been made to the manuscript below.

Reviewer’s comments:

Introduction:

Lines 64-66: A slightly pernickety point but given the literature available and described in the rest of the paragraph, it is probably not appropriate to say that understanding of the validity of remote vs in-person testing is particularly limited anymore- validity in this context is becoming reasonably well established now.

More rationale could be provided on the choice of functional tests, and this might inform a discussion point on which tests could be dropped from such a remote functional testing battery in future iterations, particularly in the context of the barriers related to set up etc.

Response: Thank you for these suggestions. We agree the statement relating to limited understanding of the validity and/or reliability of physical function tests in older adults is somewhat redundant based on recent evidence. We have therefore opted to remove this sentence altogether.

We agree with expanding on the rationale for the included tests, but felt this was best included in the methods section, where we have added the following (lines 166-170):

“The selected tests were designed to assess a broad range of physical function domains, including static balance (standing balance battery, single-leg balance test) dynamic balance and stepping speed (FSST), mobility (gait speed assessments), and lower-limb strength/strength-endurance (30-STS) and power (5-STS).”

We also agree that together with the findings regarding the test-retest reliability of the assessments, this might help to inform some improvements to the protocol that may have positive implications for the feasibility of the testing battery when performed remotely by older adults within their homes.

We have now added the following to the “Test feasibility and safety” section in the discussion (lines 582-596):

“Nevertheless, the fact that four of the seven participants who withdrew prior to the first testing session cited limited time availability (n=3) and perceived testing difficulty (n=1) suggests that shortening or simplifying the testing battery could further enhance feasibility. One strategy may be to omit tests with lower reliability or those assessing overlapping domains of physical function. For example, the single-leg balance test demonstrated poor test-retest reliability between the first two sessions (ICC = -0.29), with only moderate reliability between sessions two and three (ICC = 0.53). This is notably lower than the reliability reported in similar telehealth studies using single-leg balance tests (ICC = 0.69–0.84) (Lawford et al., 2022), and indicates this test may contribute unnecessary burden without providing robust or consistent data. Similarly, tests that capture similar constructs - such as the 4-m and 2.44-m gait speed assessments - could be streamlined, especially given the similar reliability observed for the 2.44-m fast walk (ICC = 0.73) and 4-m fast walk (ICC = 0.79). Retaining higher-performing and less redundant tests, such as the 5-STS, 30-STS, and FSST, which consistently showed moderate-to-good reliability, may streamline the testing protocol without compromising measurement quality.”

Methods:

Line 158: ‘all participants had access to these items at home’ is really a result that should be reported in the feasibility section not methods.

Response: We have removed this from methods section and added to the “Feasibility and safety of remote physical function assessments” section in the results.

Lines 188-195: were the single leg balance tests attempted for a maximum duration with no time cap?

Response: This line has been modified to:

“Participants completed a timed single-leg balance test (on their preferred leg) of a maximum duration, both with the eyes open and separately with the eyes closed.”

Lines 297-298: please specify which ICC model was used, see 10.1016/j.jcm.2016.02.012 for further details and rationale for the importance of this.

Response: We apologise for this oversight. We have now specified the ICC model used as follows (lines 299-300):

“Intraclass correlation coefficient (ICC) values (single-rating, absolute-agreement, 2-way random-effects model) were used to assess test-retest reliability.”

We believe this is the most appropriate model based on the study design and research question and aligns with the recommendations of Koo & Li, 2016.

Statistical analysis: to give a complete picture of the data, the data presented in Table 1 could be compared by repeated measures ANOVA / GLM to see if there were differences in the mean scores of each test across time.

Response: We have now analysed these data using a one-way repeated measures ANOVA and included the P-value for the main effects of time in Table 1.

Results:

Line 333: were reasons for drop-out provided, and could these be considered in the context of feasibility? On reading the qualitative results, were these drop-outs after receiving instructions for room set up? If so, this is an important consideration for feasibility?

Response: Thank you for this useful suggestion. We have now indicated the reasons for participant withdrawal as follows (lines 334-339):

“A total of 26 older adults were screened for the study, of which 26 met the eligibility criteria. Seven participants withdrew from the study prior to the first testing session and after being sent the study information and instructional video, with the remaining participants (n=19) completing all three remote functional assessments and the final interview.

Reasons for withdrawal included limited time availability (n=3), misunderstanding that the study involved a training intervention (n=2), a non-study-related injury (n=1), and the perception that the testing would be too difficult (n=1).”

We agree this informs an important discussion point regarding the feasibility of the home-based testing. As mentioned in our response to a previous comment, we have therefore added the following to the discussion section (lines 582-584):

“The present study therefore adds to our current understanding of the feasibility and safety of remote home-based physical function testing and suggests older adults with comparable physical function levels to the participants in this study can feasibly complete these assessments remotely within their home environment when monitored by a trained assessor. Nevertheless, the fact that four of the seven participants who withdrew prior to the first testing session cited limited time availability (n=3) and perceived testing difficulty (n=1) suggests that shortening or simplifying the testing battery could further enhance feasibility.”

Line 339: Weekly MVPA seems high for this age group- is that worth mentioning in the discussion?

Response: We agree this should be considered when interpreting the reliability of physical function tests in this cohort. We have incorporated this point into the discussion when mentioning how the levels of physical function in the cohort may impact upon the feasibility and/or safety of the remote physical function assessments (lines 654-655):

“The physical function levels of the older adults in this study may have influenced the feasibility and/or reliability of the physical function measures. Individuals with SPPB scores of 10 or less have a significantly higher risk of mobility disability (Vasunilashorn et al., 2009) and gait speeds of <1.0 m/s (Fielding et al., 2011) or 0.8 m/s (Cruz-Jentoft et al., 2019) are recommended for identifying those at risk of disability. Participants in this study had a median SPPB score of 9 (17/19 participants scored 10 or less; Supplementary Figure 1) and usual gait speeds of 0.73 m/s (in trial 1) for the 4 m test. While this suggests representation from older adults with reasonably poor physical function, despite the relatively high (albeit variable) weekly MVPA levels across the cohort (584 ± 606 min per week), which is often overestimated in self-reported MVPA questionnaires (Ryan et al., 2018), those with lower physical function may experience greater challenges completing physical function tests remotely.”

Lines 348-352: this feels like it is missing written results giving an objective summary of the key data in the tables and figures. Given the data is quite dry and there is a lot of it, highlighting the headlines that are later discussed is all that is needed. For readability, these written results might be best broken up to proceed the respective table or figure.

Response: Thank you for this suggestion. We have now added the following summary text under the section “Test-retest reliability of physical function assessments”(line 353-354):

“The test-retest reliability of these assessments ranged from poor (ICC<0.5; single-leg balance) to good (ICC 0.75–0.9; 5-STS, 30-STS, 2.44 m gait usual, 4 m gait fast, and FSST), and improved with repeated testing for most tests (i.e., reliability improved between testing session 2 and 3 compared to between testing session 1 and 2).”

Table 2: the negative ICC value might be indicating that there is a problem with the data, or the ICC model used is not appropriate. If these have been checked, then generally a negative ICC should not be meaningfully interpreted as the within-group variance is greater than the between-group variance, which contradicts one of the assumptions needed for ICC calculation.

Response: Thank you for this suggestion. We have double-checked both the data itself for errors and the ICC model and confirm these are both correct. Given the issues you have raised, we have included the following in the results section (lines 357-359):

“Negative ICC values were observed for the single-leg balance test between test 1 and 2, suggesting the within-group variance exceeded between-group variance. As this violates a core assumption of ICC analysis, these results should be interpreted cautiously.”

In addition, we have also included the following in the discussion section (lines 508-510):

“Of note were the negative ICC values observed for the single-leg balance test between test 1 and 2 (but not between test 2 and 3), suggesting these data should be interpreted with caution.”

Figure 1: should each panel be individually labelled and referred to in the figure legend? Also, for ease of interpretation, the axis formatting for test 1 vs test 2, and test 2 vs test 3, should be the same- this is different for 30-STS, FSST, and SPPB scores.

Response: The scaling and position of the graphs have now been improved. We have now individually labelled each graph with a letter that is also referred to in the figure legend. The equivalent changes have now been made to Figure 2 as well.

Figure 2: axis alignment is a bit off for some of the panels

Response: This has been rectified accordingly.

Layout of Qualitative Results section: it might be a quirk of the submission process, but readability would be improved if the written results for given assessment of feasibility preceded the corresponding table of quotes, i.e., summary of participant’s enablers followed by table of participant’s enabler quotes, then summary of assessor’s enablers followed by table of assessor’s enabler quotes.

Response: Thank you for this suggestion. We have re-ordered the qualitative results sections accordingly, with the written results for barriers (for both participants and assessors) followed by the relevant tables with quotes, and the same applied for the enablers.

Discussion:

- Given the interview was after third assessment only, is there risk of recency bias, for example when considering technical difficulty? After three sessions, technical issues of the first session may be downplayed.

Response: We agree the timing of the interview in relation to the testing sessions may have influenced participants’ perceptions the sessions, including aspects related to technical difficulty. We have included the following line in the limitations section (lines 667-669):

“Conducting interviews only after the third testing session may have introduced recency bias, where participants' responses were disproportionately influenced by their most recent experience, such as any technical difficulties encountered, which, by that stage, had been largely resolved for all participants.”

- Possibly linking back to a comment on the introduction regarding rationale for inclusion of each test, providing pragmatic suggestions for how these data might inform future implementation (alluded to in lines 585-587) would be an interesting and useful addition to the manuscript.

Response: Thank you for this suggestion. We believe the changes made to the “Test feasibility and safety” section in response to previous comments have now satisfied this query.

- A limitation that ought to be mentioned was that no comparison of reliability between assessors was made, though that would have required a larger sample size and different ICC model.

Response: We agree this is an important limitation to mention. We have included the following line in the limitations section (lines 665-667):

“We also did not evaluate inter-rater reliability, although this would have required both a larger sample and different ICC methodology.”

---

## [Decision Letter · Decision Letter 1]

3 Sep 2025

Remote physical function testing in older adults: a mixed methods study exploring test reliability, feasibility, and perceptions of participants and assessors.

PONE-D-25-19949R1

Dear Dr. Jackson J Fyfe,

We're pleased to inform you that your manuscript has been judged scientifically suitable for publication and will be formally accepted for publication once it meets all outstanding technical requirements.

Kind regards,

Jindong Chang, Ph.D.

Academic Editor

PLOS ONE

Additional Editor Comments (optional):

Reviewer #1:

Reviewers' comments:

Reviewer's Responses to Questions

**Comments to the Author**

1. If the authors have adequately addressed your comments raised in a previous round of review and you feel that this manuscript is now acceptable for publication, you may indicate that here to bypass the “Comments to the Author” section, enter your conflict of interest statement in the “Confidential to Editor” section, and submit your "Accept" recommendation.

Reviewer #1: All comments have been addressed

2. Is the manuscript technically sound, and do the data support the conclusions?

Reviewer #1: Yes

3. Has the statistical analysis been performed appropriately and rigorously? 

Reviewer #1: Yes

4. Have the authors made all data underlying the findings in their manuscript fully available?

Reviewer #1: Yes

5. Is the manuscript presented in an intelligible fashion and written in standard English?

Reviewer #1: Yes

6. Review Comments to the Author

Reviewer #1: The authors have thoroughly addressed all my comments, the additional text is clear and adds further insight to the manuscript.

7. PLOS authors have the option to publish the peer review history of their article (what does this mean? ). If published, this will include your full peer review and any attached files.

**Do you want your identity to be public for this peer review?** For information about this choice, including consent withdrawal, please see our Privacy Policy .

Reviewer #1: No

---

## [Editor Report · Acceptance letter]

PONE-D-25-19949R1

PLOS ONE

Dear Dr. Fyfe,

I'm pleased to inform you that your manuscript has been deemed suitable for publication in PLOS ONE. Congratulations! Your manuscript is now being handed over to our production team.

Kind regards,

on behalf of

Dr. Jindong Chang

Academic Editor

PLOS ONE